# RECURRENT HIDDEN SEMI-MARKOV MODEL

**Hanjun Dai[1], Bo Dai[1], Yan-Ming Zhang[2], Shuang Li[1], Le Song[1]**

[1] Georgia Institute of Technology
{hanjundai, bodai, sli370}@gatech.edu, lsong@cc.gatech.edu
[2] National Laboratory of Pattern Recognition, Chinese Academy of Sciences
ymzhang@nlpr.ia.ac.cn

## ABSTRACT

Segmentation and labeling of high dimensional time series data has wide applications in behavior understanding and medical diagnosis. Due to the difficulty of obtaining a large amount the label information, realizing this objective in an unsupervised way is highly desirable. Hidden Semi-Markov Model (HSMM) is a classical tool for this problem. However, existing HSMM and its variants typically make strong generative assumptions on the observations within each segment, thus their ability to capture the nonlinear and complex dynamics within each segment is limited. To address this limitation, we propose to incorporate the Recurrent Neural Network (RNN) as the generative process of each segment, resulting the Recurrent HSMM (R-HSMM). To accelerate the inference while preserving accuracy, we designed a structure encoding function to mimic the exact inference. By generalizing the penalty method to distribution space, we are able to train the model and the encoding function simultaneously. We also demonstrate that the R-HSMM significantly outperforms the previous state-of-the-art on both the synthetic and real-world datasets.

## 1 INTRODUCTION

Segmentation and labeling of time series data is an important problem in machine learning and signal processing. Given a sequence of observations $\{x_1, x_2, \ldots, x_T\}$, we want to divide the $T$ observations into several segments and label each segment simultaneously, where each segment consists of consecutive observations. The supervised sequence segmentation or labeling techniques have been well studied in recent decades (Sutskever et al., 2014; Kong et al., 2015; Chen et al., 2015). However, for complicated signals, like human activity sensor data, accurately annotating the segmentation boundary or the activity type would be prohibitive. Therefore, it is urgent to develop unsupervised algorithms that can jointly learn segmentation and labeling information directly from the data without supervisions. Figure 1 provides an illustration which we are focus on.

The Hidden Semi-Markov Model (HSMM) (Murphy, 2002) is a powerful model for such task. It eliminates the implicit geometric duration distribution assumptions in HMM (Yu, 2010), thus allows the state to transit in a non-Markovian way. Most of the HSMM variants make strong parametric assumptions on the observation model (Rabiner, 1989; Johnson & Willsky, 2013; Yu, 2010). This makes the learning and inference simple, but ignores the nonlinear and long-range dependency within a segment. Take the human activity signals as an example. The movements a person performs at a certain time step would rely heavily on the previous movements, like the interleaving actions of left hand and right hand in swimming, or more complicated dependency like shooting after jumping in playing basketball. Some models have been proposed to tackle this problem (Ghahramani & Hinton, 2000; Fox et al., 2009; Linderman et al., 2016), but are limited in linear case.

Since people have justified RNN's ability in modeling nonlinear and complicated dependencies (Sutskever et al., 2014; Du et al., 2016), we introduce the recurrent neural emission model into HSMM for capturing various dependencies within each segment to address such issue. However, the flexibility of recurrent neural model comes with prices: it makes the exact Expectation-Maximization (EM) algorithm computationally too expensive.

To speed up the learning and inference, we exploit the variational encoder (VAE) framework (Kingma & Welling, 2013). Specifically, we propose to use bidirectional RNN (bi-RNN) encoder. Such

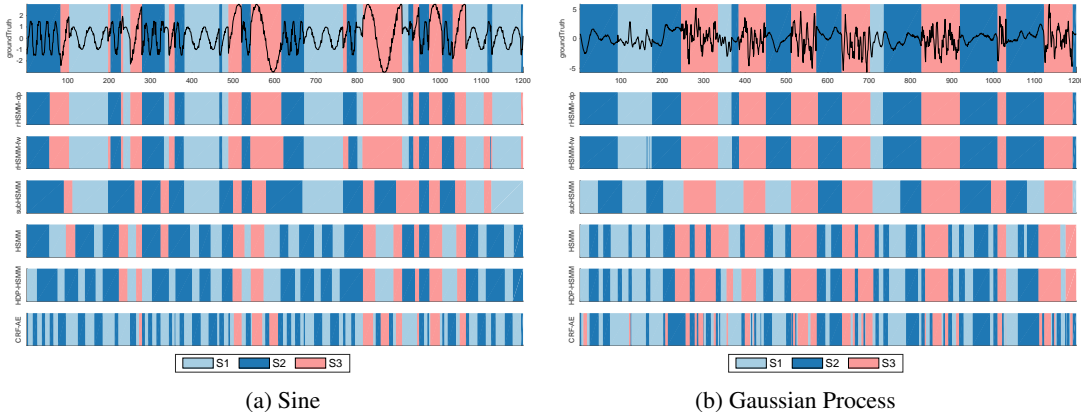

(a) Sine (b) Gaussian Process

Figure 1: Synthetic experiment results. Different background colors represent the segmentations with different labels. In the top row, the black curve shows the raw signal. (a) The Sine data set is generated by a HSMM with 3 hidden states, where each one has a corresponding sine function; (b) Similar to 1a, but the segments are generated from Gaussian processes with different kernel functions. The first two rows are our algorithms which almost exact locate every segment.

architecture will mimic the forward-backward algorithm, and hence is expected to capture similar information as in exact posterior calculation.

It should be emphasized that due to the *discrete* nature of the latent variables in our model, the algorithm proposed in Kingma & Welling (2013) and its extension on time-series models (Gao et al., 2016; Krishnan et al., 2015) are not directly applicable. There are plenty of work proposed based on stochastic neuron (Tang & Salakhutdinov, 2013; Bengio et al., 2013; Mnih & Gregor, 2014; Raiko et al., 2014; Gu et al., 2015; Chung et al., 2016) to remedy such issue. However, none of these off-the-shelf methods are easy to achieve good performance according to our experiment: the hundreds or thousands layers of stochastic neuron (which is equal to the length of sequence), together with the switching generative RNN, make the encoding function very sensitive, and thus, extremely difficult to train fully on unsupervised setting. We propose a solution, *stochastic distributional penalty method*, which introduces auxiliary distributions to separate the decoding R-HSMM and encoding bi-RNN in training procedure, and thus, reduces the learning difficulty for each component. This novel algorithm is general enough and can be applied to other VAE with discrete latent variables, which can be of independent interest. We emphasize that the proposed algorithm is maximizing *exact* the nagative Helmholtz variational free energy. It is different from Johnson et al. (2016) in which a lower bound of the variational free energy is proposed as the surrogate to be maximized for convenience.

We experimentally justified our algorithm on the synthetic datasets and three real-world datasets, namely the segmentation tasks for human activity, fruit fly behavior and heart sound records. The R-HSMM with Viterbi exact inference *significantly outperforms* basic HSMM and its variants, demonstrating the generative model is indeed flexible. Moreover, the trained bi-RNN encoder also achieve similar *state-of-the-art performances* to the exact inference, but with *400 times faster* inference speed, showing the proposed structured encoding function is able to mimic the exact inference efficiently.

## 2 MODEL ARCHITECTURE

Given a sequence $\boldsymbol{x} = [x_1, x_2, \ldots, x_{|\boldsymbol{x}|}]$, where $x_t \in \mathbb{R}^m$ is an $m$ dimensional observation at time $t$, our goal is to divide the sequence into meaningful segments. Thus, each observation $x_t$ will have the corresponding label $z_t \in \mathbb{Z}$, where $\mathbb{Z} = \{1, 2, \ldots, K\}$ is a finite discrete label set and $K$ is predefined. The label sequence $\boldsymbol{z} = [z_1, z_2, \ldots, z_{|\boldsymbol{x}|}]$ should have the same length of $\boldsymbol{x}$.

Besides labels, HSMM will associate each position $t$ with additional variable $d_t \in \mathbb{D} = \{1, 2, \ldots, D\}$, where $d_t$ is known as duration variable and $D$ is the maximum possible duration. The duration variable can control the number of steps the current hidden state will remain. We use $\boldsymbol{d}$ to denote the duration sequence. We also use notation $\boldsymbol{x}_{t_1:t_2}$ to denote the substring $[x_{t_1}, x_{t_1+1}, \ldots, x_{t_2}]$ of $\boldsymbol{x}$. Without ambiguity, we use $z$ as a segment label, and $d$ as the duration.

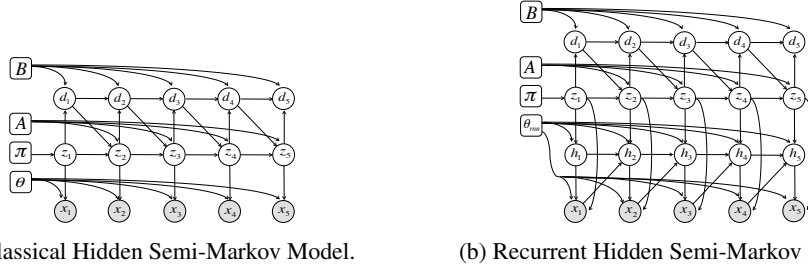

(a) Classical Hidden Semi-Markov Model.   (b) Recurrent Hidden Semi-Markov Model.

Figure 2: Graphical models of HSMM and R-HSMM. Different from classical HSMM, the R-HSMM has two-level emission structure with recurrent dependency.

In this paper, we focus on one of the variants of HSMM, namely the explicit duration HMM (EDHMM) (Rabiner, 1989), and use Decreasing Count Variables (Chiappa, 2014) for the notation.

**Explicit Duration Hidden Markov Model.** Similar to HMM, this model treats the pair of $(z, d)$ as 'macro hidden state'. The probability of initial macro state is defined as $P(z, d) = P(z)P(d|z)$. We use the notation $\pi_z \triangleq P(z)$ and $P(d|z) \triangleq B_{z,d}$ to parametrize the initial probability and duration probability, respectively. $A_{i,j} \triangleq P(z_t = i|z_{t-1} = j, d_{t-1} = 1)$ is the state transition probability on the segment boundary. Here $\pi \in \mathbb{R}^K$ is in $K$-dimensional simplex. For each hidden state $z$, the corresponding rows $B_{z,:}$ and $A_{z,:}$ are also in probability simplex. Here we assume the multinomial distribution for $P(d|z)$.

In EDHMM, the transition probability of macro hidden state $P(z_t, d_t|z_{t-1}, d_{t-1})$ is decomposed by $P(z_t|z_{t-1}, d_{t-1})P(d_t|z_t, d_{t-1})$ and thus can be defined as:

$$P(z_t|z_{t-1}, d_{t-1}) = \begin{cases} A_{z_{t-1}, z_t} & \text{if } d_{t-1} = 1 \\ \mathbb{I}_{(z_t = z_{t-1})} & \text{if } d_{t-1} > 1 \end{cases}; \quad P(d_t|z_t, d_{t-1}) = \begin{cases} B_{z_t, d_t} & \text{if } d_{t-1} = 1 \\ \mathbb{I}_{(d_t = d_{t-1} - 1)} & \text{if } d_{t-1} > 1 \end{cases}. \quad (1)$$

The graphical model is shown in Figure 2a.

**Recurrent Hidden Semi-Markov Model.** For the simplicity of explanation, we focus our algorithm on the single sequence first. It is straightforward to apply the algorithm for dataset that has multiple sequences. Given the parameters $\{\pi, A, B\}$, the log-likelihood of a single observation sequence $\boldsymbol{x}$ can be written as below,

$$\mathcal{L}(\boldsymbol{x}) = \log \sum_{\boldsymbol{z}, \boldsymbol{d}} \pi_{z_1} B_{z_1, d_1} \prod_{t=2}^{|\boldsymbol{x}|} P(z_t|z_{t-1}, d_{t-1}) P(d_t|z_t, d_{t-1}) P(\boldsymbol{x}|\boldsymbol{z}, \boldsymbol{d}), \quad (2)$$

where $P(\boldsymbol{x}|\boldsymbol{z}, \boldsymbol{d})$ is the emission probability. To define $P(\boldsymbol{x}|\boldsymbol{z}, \boldsymbol{d})$, we further denote the sequence variable $\boldsymbol{s} = [s_1, s_2, \ldots, s_{|\boldsymbol{s}|}]$ to be the switching position (or the beginning) of segments. Thus $s_1 = 1$, and $s_i = s_{i-1} + d_{s_{i-1}}$ and $|\boldsymbol{s}|$ is the number of segments. Traditional HSMM assumes $P(\boldsymbol{x}|\boldsymbol{z}, \boldsymbol{d}) = \prod_{t=1}^{|\boldsymbol{x}|} P(x_t|z_t)$, which ignores the dependency and some degree of dynamics exhibited in each segment. While in this paper, we use RNN as the generative model to capture the dependent emission probability. Specifically, for the $i$-th segment starting from position $s_i$, the corresponding generative process is

$$P(\boldsymbol{x}_{s_i:s_i+d_{s_i}-1}|z_{s_i}, d_{s_i}) = \prod_{t=s_i}^{s_i+d_{s_i}-1} P(x_t|x_{s_i:t-1}, z_{s_i}) = \prod_{t=s_i}^{s_i+d_{s_i}-1} P(x_t|h_t, z_{s_i}) \quad (3)$$

where we assume that the dependency of history before time step $j$ can be captured by a vector $h_j \in \mathbb{R}^h$. As in RNN, we use a recurrent equation to formulate the history vector,

$$h_t = \sigma(W^{(z_{s_i})} x_{t-1} + V^{(z_{s_i})} h_{t-1} + b^{(z_{s_i})}). \quad (4)$$

Finally, in this model, $P(\boldsymbol{x}|\boldsymbol{z}, \boldsymbol{d}) = \prod_{i=1}^{|\boldsymbol{s}|} P(\boldsymbol{x}_{s_i:s_i+d_{s_i}-1}|z_{s_i}, d_{s_i})$ is computed by the product of generative probabilities for each segment. In Eq. 4, $W \in \mathbb{R}^{m \times h}$ is a weight matrix capturing the last observation $x_{t-1}$, and $V \in \mathbb{R}^{h \times h}$ is for the propagation of history $h_{t-1}$. The $b$ is a bias term. The superscript $z_{s_i}$ indexes the RNN we used for the corresponding segment. The segments with different labels are generated using different RNNs. So we should maintain $K$ RNNs. $\sigma(\cdot)$ is a nonlinear activation function. We use tanh in our experiments.

At the time step $t$, we assume a diagonal multivariate gaussian distribution over the conditional likelihood, where the mean and covariance matrix are the output of RNN, *i.e.*,

$$P(x_t|h_t, z_{s_i}) \sim \mathcal{N}(x_t; \mu = U_\mu^{(z_{s_i})} h_t + b_\mu^{(z_{s_i})}, \Sigma = \text{Diag}(\exp(U_\Sigma^{(z_{s_i})} h_t + b_\Sigma^{(z_{s_i})}))) \qquad (5)$$

The matrices $U_\mu, U_\Sigma \in \mathbb{R}^{m \times h}$ are used for parametrizing the mean and covariance at each time step $j$, given the history information. $b_\mu, b_\Sigma \in \mathbb{R}^m$ are bias terms. For simplicity, let's use $\theta_{rnn} = \{\theta_{rnn}^{(1)}, \theta_{rnn}^{(2)}, \dots, \theta_{rnn}^{(K)}\}$ to denote the collection of parameters in each RNN. On the boundary case, *i.e.*, the starting point of each segment, we simply set $h_t = 0$, which can be viewed as the setting according to the prior knowledge (bias terms) of RNN.

The above formulation indicates that the generative model $P(x_t|h_t, z_{s_i})$ depends not only on the last step observation $x_{t-1}$, but also the last hidden state $h_{t-1}$, which is together captured in Eq. 4. In summary, we denote all the parameters in the proposed R-HSMM as $\theta = \{\pi, A, B, \theta_{rnn}\}$. The corresponding graphical model is shown in Figure 2b.

## 3 SEQUENTIAL VARIATIONAL AUTOENCODER

To obtain the posterior or MAP in the proposed R-HSMM, the classical forward-backward algorithm or Viterbi algorithm needs to solve one dynamic programming *per sample*, which makes the inference costly, especially for the long sequence with thousands of timestamps. So instead, we treat the Bayesian inference from optimization perspective, and obtain the posterior by maximizing the negative Helmholtz variational free energy (Williams, 1980; Zellner, 1988; Dai et al., 2016),

$$\max_{Q(\boldsymbol{z}, \boldsymbol{d}|\boldsymbol{x}) \in \mathcal{P}} \mathcal{L}_Q^\theta(x) := \mathbb{E}_{Q(\boldsymbol{z}, \boldsymbol{d}|\boldsymbol{x})} \left[\log P_\theta(\boldsymbol{x}, \boldsymbol{z}, \boldsymbol{d}) - \log Q(\boldsymbol{z}, \boldsymbol{d}|\boldsymbol{x})\right], \qquad (6)$$

over the space of all valid densities $\mathcal{P}$. To make the optimization (6) tractable, the variational autoencoder restricts the feasible sets to be some parametrized density $Q_\psi$, which can be executed efficiently comparing to the forward-backward algorithm or Viterbi algorithm. However, such restriction will introduce extra approximation error. To reduce the approximation error, we use a structured model, *i.e.*, bidirectional RNN, to mimic the dynamic programming in forward-backward algorithm. Specifically, in the forward-backward algorithm, the forward message $\alpha_t(z_t, d_t)$ and backward message $\beta_t(z_t, d_t)$ can be computed recursively, and marginal posterior at position $t$ depends on both $\alpha_t(z_t, d_t)$ and $\beta_t(z_t, d_t)$. Similarly, in bi-RNN we embed the posterior message with RNN's latent vector, and marginal posterior is obtained from the latent vectors of two RNNs at the same time step $t$. Let $\psi = \{\psi_{\overrightarrow{\text{RNN}_1}}, \psi_{\overleftarrow{\text{RNN}_2}}, W_z \in \mathbb{R}^{h \times K}, W_d \in \mathbb{R}^{h \times D}\}$ be the parameters of bi-RNN encoder, the $Q_\psi$ is decomposed as:

$$Q_\psi(\boldsymbol{z}, \boldsymbol{d}|\boldsymbol{x}) = Q(z_1|h_1; \psi)Q(d_1|z_1, h_1; \psi) \prod_{t=2}^{|\boldsymbol{x}|} Q(z_t|d_{t-1}, h_t; \psi)Q(d_t|z_t, d_{t-1}, h_t; \psi), \qquad (7)$$

where $h_t = [\overrightarrow{\text{RNN}_1}(\boldsymbol{x}_{1:t}), \overleftarrow{\text{RNN}_2}(\boldsymbol{x}_{t:|\boldsymbol{x}|})]$ is computed by bi-RNN. We use multinomial distributions $Q(z_t|h_t; \psi) = \mathcal{M}(\text{softmax}(W_z^\top h_t))$ and $Q(d_t|z_t, h_t; \psi) = \mathcal{M}(\text{softmax}(W_d^\top h_t))$. The dependency over $d_{t-1}$ ensures that the generated segmentation $(\boldsymbol{z}, \boldsymbol{d})$ is valid according to Eq. 1. For example, if we sampled duration $d_{t-1} > 1$ from $Q_\psi$ at time $t - 1$, then $d_t$ and $z_t$ should be deterministic. In our experiment, we use LSTM (Hochreiter & Schmidhuber, 1997) as the recursive units in bi-RNN.

Since with any fixed $\theta$, the negative Helmholtz variational free energy is indeed the *lower* bound of the marginal likelihood, *i.e.*,

$$\log P_\theta(\boldsymbol{x}) \geq \mathcal{L}(\theta, \psi; \boldsymbol{x}) := \mathbb{E}_{Q_\psi(\boldsymbol{z}, \boldsymbol{d}|\boldsymbol{x})}[\log P_\theta(\boldsymbol{x}, \boldsymbol{z}, \boldsymbol{d}) - \log Q_\psi(\boldsymbol{z}, \boldsymbol{d}|\boldsymbol{x})], \qquad (8)$$

therefore, we can treat it as a surrogate of the marginal log-likelihood and learn $\theta$ jointly with approximate inference, *i.e.*,

$$\max_{\theta, \psi} \frac{1}{N} \sum_{n=1}^{N} \mathcal{L}(\theta, \psi; \boldsymbol{x}^{(n)}) \qquad (9)$$

It should be emphasized that due to the *discrete* nature of latent variables in our model, the algorithm proposed in Kingma & Welling (2013) is not directly applicable, and its extension with stochastic neuron reparametrization (Bengio et al., 2013; Raiko et al., 2014; Gu et al., 2015; Chung et al., 2016) cannot provide satisfied results for our model according to our experiments. Therefore, we extend the penalty method to distribution space to solve optimization (9).

---

**Algorithm 1 Learning sequential VAE with stochastic distributional penalty method**

---

1: **Input:** sequences $\{\boldsymbol{x}^{(n)}\}_{n=1}^N$
2: Randomly initialize $\psi^{(0)}$ and $\theta^0 = \{\pi^0, A^0, B^0, \theta_{rnn}^0\}$
3: **for** $\lambda = 0, \dots, \infty$ **do**
4:     **for** $t = 0$ **to** $T$ **do**
5:         Sample $\{\boldsymbol{x}^{(n)}\}_{n=1}^M$ uniformly from dataset with mini-batch size $M$.
6:         Get $\{\boldsymbol{z}^{(n)}, \boldsymbol{d}^{(n)}\}_{n=1}^M$ with $\theta^t$ by dynamic programming in (13).
7:         Update $\pi^{t+1}, A^{t+1}, B^{t+1}$ using rule (16).
8:         $\theta_{rnn}^{t+1} = \theta_{rnn}^t, -\gamma_t \frac{1}{M} \sum_{n=1}^M \nabla_{\theta_{rnn}^t} \tilde{\mathcal{L}}_\lambda(\theta, \psi|\boldsymbol{x}^{(n)})$
9:         $\psi^{t+1} = \psi^t - \eta_t \frac{1}{M} \sum_{n=1}^M \nabla_{\psi^t} \tilde{\mathcal{L}}_\lambda(\theta, \psi|\boldsymbol{x}^{(n)})$     ▷ bi-rnn sequence to sequence learning
10:     **end for**
11: **end for**

---

## 4 LEARNING VIA STOCHASTIC DISTRIBUTIONAL PENALTY METHOD

As we discussed, learning the sequential VAE with stochastic neuron reparametrization in unsupervised setting is extremely difficult, and none the off-the-shelf techniques can provide satisfied results. In this section, we introduce *auxiliary distribution* into (9) and generalize the penalty method Bertsekas (1999) to distribution space.

Specifically, we first introduce an auxiliary distribution $\tilde{Q}(\boldsymbol{z}, \boldsymbol{d}|\boldsymbol{x})$ for each $\boldsymbol{x}$ and reformulate the optimization (9) as

$$\max_{\theta, \psi, \{\tilde{Q}(\boldsymbol{z}, \boldsymbol{d}|\boldsymbol{x}^{(n)})\}_{n=1}^N} \frac{1}{N} \sum_{n=1}^N \mathbb{E}_{\tilde{Q}(\boldsymbol{z}, \boldsymbol{d}|\boldsymbol{x}^{(n)})} \left[ \log P_\theta(\boldsymbol{x}^{(n)}, \boldsymbol{z}, \boldsymbol{d}) - \log \tilde{Q}(\boldsymbol{z}, \boldsymbol{d}|\boldsymbol{x}^{(n)}) \right], \quad (10)$$

$$\text{s.t.} \quad KL\left(\tilde{Q}(\boldsymbol{z}, \boldsymbol{d}|\boldsymbol{x}^{(n)}) || Q_\psi(\boldsymbol{z}, \boldsymbol{d}|\boldsymbol{x}^{(n)})\right) = 0, \ \forall \boldsymbol{x}^{(n)}, n = 1, \dots, N.$$

We enforce the introduced $\tilde{Q}(\boldsymbol{z}, \boldsymbol{d}|\boldsymbol{x})$ equals to $Q_\psi(\boldsymbol{z}, \boldsymbol{d}|\boldsymbol{x})$ in term of $KL$-divergence, so that the optimization problems (9) and (10) are equivalent. Because of the non-negativity of $KL$-divergence, itself can be viewed as the penalty function, we arrive the alternative formulation of (10) as

$$\max_{\theta, \psi, \{\tilde{Q}(\boldsymbol{z}, \boldsymbol{d}|\boldsymbol{x}^{(n)})\}_{n=1}^N} \frac{1}{N} \sum_{n=1}^N \tilde{\mathcal{L}}_\lambda(\theta, \psi|\boldsymbol{x}^{(n)}), \quad (11)$$

where

$$\tilde{\mathcal{L}}_\lambda(\theta, \psi|\boldsymbol{x}) = \mathbb{E}_{\tilde{Q}(\boldsymbol{z}, \boldsymbol{d})} \left[ \log P_\theta(\boldsymbol{x}, \boldsymbol{z}, \boldsymbol{d}) - \log \tilde{Q}(\boldsymbol{z}, \boldsymbol{d}|\boldsymbol{x}_i) \right] - \lambda KL\left(\tilde{Q}(\boldsymbol{z}, \boldsymbol{d}|\boldsymbol{x}) || Q_\psi(\boldsymbol{z}, \boldsymbol{d}|\boldsymbol{x})\right)$$

and $\lambda \geq 0$. Obviously, as $\lambda \to \infty$, $KL(\tilde{Q}(\boldsymbol{z}, \boldsymbol{d}|\boldsymbol{x}) || Q_\psi(\boldsymbol{z}, \boldsymbol{d}|\boldsymbol{x}))$ must be 0, otherwise the $\tilde{\mathcal{L}}_\infty(\theta, \psi|\boldsymbol{x})$ will be $-\infty$ for arbitrary $\theta, \psi$. Therefore, the optimization (11) will be equivalent to problem (10). Following the penalty method, we can learn the model with $\lambda$ increasing from 0 to $\infty$ gradually. The entire algorithm is described in Algorithm 1. Practically, we can set $\lambda = \{0, \infty\}$ and do no need the gradually incremental, while still achieve satisfied performance. For each fixed $\lambda$, we optimize $\tilde{Q}$ and the parameters $\theta, \psi$ alternatively. To handle the expectation in the optimization, we will exploit the stochastic gradient descent. The update rules for $\theta, \psi$ and $\tilde{Q}$ derived below.

### 4.1 UPDATING $\tilde{Q}$

In fact, fix $\lambda$, $Q_\psi$ and $P_\theta$ in optimization (11), the optimal solution $\tilde{Q}^*(\boldsymbol{z}, \boldsymbol{d}|\boldsymbol{x})$ for each $\boldsymbol{x}$ has closed-form.

**Theorem 1** *Given fixed $\lambda$, $Q_\psi$ and $P_\theta$, $\tilde{Q}^*(\boldsymbol{z}, \boldsymbol{d}|\boldsymbol{x}) \propto Q_\psi(\boldsymbol{z}, \boldsymbol{d}|\boldsymbol{x})^{\frac{\lambda}{1+\lambda}} P_\theta(\boldsymbol{x}, \boldsymbol{z}, \boldsymbol{d})^{\frac{1}{1+\lambda}}$ achieves the optimum in (11).*

**Proof** The proof is straightforward. Take the functional derivative of $\tilde{\mathcal{L}}$ w.r.t. $\tilde{Q}$ and set it to zeros,

$$\nabla_{\tilde{Q}} \tilde{\mathcal{L}} = \log P_\theta(\boldsymbol{x}, \boldsymbol{z}, \boldsymbol{d}) + \lambda \log Q(\boldsymbol{z}, \boldsymbol{d}|\boldsymbol{x}) - (1 + \lambda) \log \tilde{Q}(\boldsymbol{z}, \boldsymbol{d}|\boldsymbol{x}) = 0,$$

we obtain $\frac{1}{1+\lambda} \log P_\theta(\boldsymbol{x}, \boldsymbol{z}, \boldsymbol{d}) + \frac{\lambda}{1+\lambda} \log Q(\boldsymbol{z}, \boldsymbol{d}|\boldsymbol{x}) = \log \tilde{Q}(\boldsymbol{z}, \boldsymbol{d}|\boldsymbol{x})$. Take exponential on both sides, we achieve the conclusion. ∎

In fact, because we are using the stochastic gradient for updating $\theta$ and $\psi$ later, $\tilde{Q}^*(\boldsymbol{z}, \boldsymbol{d}|\boldsymbol{x})$ is never explicitly computed and only samples from it are required. Recall the fact that $Q_\psi(\boldsymbol{z}, \boldsymbol{d}|\boldsymbol{x})$ has a nice decomposition 7, we can multiply its factors into each recursion step and still get the same complexity

as original Viterbi algorithm for MAP or sampling. Specifically, let's define $\alpha_t(j, r)$ to be the best joint log probability of prefix $x_{1:t}$ and its corresponding segmentation which has the last segment with label $j$ and duration $r$, *i.e.*,

$$\alpha_t(j, r) \triangleq \max_{\boldsymbol{z}_{1:t}, \boldsymbol{d}_{1:t}} \log \tilde{Q}(\boldsymbol{z}_{1:t}, \boldsymbol{d}_{1:t} | \boldsymbol{x}_{1:t}), \text{ s.t. } z_t = j, d_t = d_{t-r} = 1, d_{t-r+1} = r \quad (12)$$

here $t \in \{1, 2, \ldots, |\boldsymbol{x}|\}, j \in \mathbb{Z}, r \in \mathbb{D}$. Then we can recursively compute the entries in $\alpha$ as below:

$$\alpha_t(j, r) = \begin{cases} \alpha_{t-1}(j, r-1) + \frac{1}{1+\lambda} \log(\frac{B_{j,r}}{B_{j,r-1}} P(x_t | x_{t-r+1:t-1}, z = j)) & r > 1, t > 1 \\ \quad + \frac{\lambda}{1+\lambda} \log \frac{Q_\psi(d_{t-r+1}=r|z=j, \boldsymbol{x})}{Q_\psi(d_{t-r+1}=r-1|z=j, \boldsymbol{x})}; & \\ \max_{i \in \mathbb{Z} \setminus j} \max_{r' \in \mathbb{D}} \alpha_{t-1}(i, r') + \frac{1}{1+\lambda} \log(A_{i,j} B_{j,1} P(x_t | z = j)) & r = 1, t > 1 \\ \quad + \frac{\lambda}{1+\lambda} \log Q_\psi(z_{t-r+1} = j, d_{t-r+1} = r | \boldsymbol{x}); & \\ \frac{\lambda}{1+\lambda} \log Q_\psi(z_1 = j, d_1 = r | \boldsymbol{x}) + \frac{1}{1+\lambda} \log(\pi_j B_{j,1} P(x_1 | z = j)); & r = 1, t = 1 \\ 0. & \text{otherwise} \end{cases}$$
$$(13)$$

To construct the MAP solution, we also need to keep a back-tracing array $\beta_t(j, r)$ that records the transition path from $\alpha_{t-1}(i, r')$ to $\alpha_t(j, r)$. The sampling from $\tilde{Q}(\boldsymbol{z}, \boldsymbol{d} | \boldsymbol{x})$ also can be completed with almost the same style forwards filtering backwards sampling algorithm, except replacing the max-operator by sum-operator in $\alpha$ propagation Murphy (2012).

Without considering the complexity of computing emission probabilities, the dynamic programming needs time complexity $\mathcal{O}\left(|\boldsymbol{x}|K^2 + |\boldsymbol{x}|KD\right)$ (Yu & Kobayashi, 2003) and $\mathcal{O}(|\boldsymbol{x}|K)$ memory. We explain the details of optimizing the time and memory requirements in Appendix A.

**Remark:** When $\lambda = \infty$, the $\tilde{Q}(\boldsymbol{z}, \boldsymbol{d} | \boldsymbol{x})$ will be exactly $Q_\psi(\boldsymbol{z}, \boldsymbol{d} | \boldsymbol{x})$ and the algorithm will reduce to directly working on $Q_\psi(\boldsymbol{z}, \boldsymbol{d} | \boldsymbol{x})$ without the effect from $P_\theta(\boldsymbol{x}, \boldsymbol{z}, \boldsymbol{d})$. Therefore, it is equivalent to obtaining MAP or sampling of the latent variables $\boldsymbol{z}, \boldsymbol{d}$ from $Q_\psi(\boldsymbol{z}, \boldsymbol{d} | \boldsymbol{x})$, whose cost is $\mathcal{O}(|\boldsymbol{x}|K)$. In practical, to further accelerate the computation, we can follow such strategy to generate samples when $\lambda$ is already large enough, and thus, the effect of $P_\theta(\boldsymbol{x}, \boldsymbol{z}, \boldsymbol{d})$ is negligible.

## 4.2 UPDATING $\theta$ AND $\psi$

With the fixed $\tilde{Q}(\boldsymbol{z}, \boldsymbol{d} | \boldsymbol{x})$, we can update the $\theta$ and $\psi$ by exploiting stochastic gradient descent algorithm to avoid scanning the whole training set. Sample a mini-batch of sequences $\{\boldsymbol{x}^n\}_{n=1}^M$ with size $M \ll N$, we proceed to update $\{\theta, \psi\}$ by optimizing the Monte Carlo approximation of (11),

$$\max_{\theta, \psi} \frac{1}{M} \sum_{n=1}^M \log P_\theta(\boldsymbol{x}^{(n)}, \boldsymbol{z}^{(n)}, \boldsymbol{d}^{(n)}) + \lambda \log Q_\psi(\boldsymbol{z}^{(n)}, \boldsymbol{d}^{(n)} | \boldsymbol{x}^{(n)}), \quad (14)$$

where $\{\boldsymbol{z}^{(n)}, \boldsymbol{d}^{(n)}\}$ is the MAP or a sample of $\tilde{Q}(\boldsymbol{z}, \boldsymbol{d} | \boldsymbol{x}^{(n)})$. Note that the two parts related to $\theta$ and $\psi$ are separated now, we can optimize them easily.

**Update $\theta$:** Finding parameters to maximize the likelihood needs to solve the constrained optimization shown below

$$\max_\theta \frac{1}{M} \sum_{n=1}^M \left( \log \pi_{z_1^{(n)}} + \sum_{i=2}^{|\boldsymbol{s}|} \log A_{z_{s_{i-1}}^{(n)}, z_{s_i}^{(n)}} + \sum_{i=1}^{|\boldsymbol{s}|} B_{z_{s_i}^{(n)}, d_{s_i}^{(n)}} + \sum_{j=s_i}^{s_i + d_{s_i}^{(n)} - 1} \log P(x_j^{(n)} | h_j^{(n)}, z_{s_i}^{(n)}; \theta_{rnn}) \right)$$
$$(15)$$

where $\{\pi, A, B\}$ are constrained to be valid probability distribution. We use stochastic gradient descent to update $\theta_{rnn}$ in totally $K$ RNNs. For parameters $\pi, A, B$ which are restricted to simplex, the stochastic gradient update will involve extra projection step. To avoid such operation which may be costly, we propose the closed-form update rule derived by Lagrangian,

$$\pi_i = \frac{\sum_{n=1}^M \mathbb{I}(z_1^{(n)} = i)}{m}, \quad A_{i,j} = \frac{\sum_{n=1}^M \sum_{t=1}^{|\boldsymbol{s}^{(n)}|-1} \mathbb{I}(z_{s_t}^{(n)} = i \text{ and } z_{s_{t+1}}^{(n)} = j)}{\sum_{n=1}^M |\boldsymbol{s}^{(n)}| - M} \quad (16)$$

$$B_{j,r} = \frac{\sum_{n=1}^M \sum_{t=1}^{|\boldsymbol{s}^{(n)}|} \mathbb{I}(d_{s_t}^{(n)} = r \text{ and } z_{s_t}^{(n)} = j)}{\sum_{n=1}^M |\boldsymbol{s}^{(n)}|}$$

Since we already have the segmentation solution, the total number of samples used for training is equal to the number of observations in dataset. The different RNNs use different parameters, and train on different parts of observations. This makes it easy for parallelized training.

**Update** $\psi$: Given fixed $\lambda$, $\log Q_\psi(\boldsymbol{z}^{(n)}, \boldsymbol{d}^{(n)}|\boldsymbol{x}^{(n)})$ is essentially the sequence to sequence likelihood, where the input sequence is $\boldsymbol{x}$ and output sequence is $\{\boldsymbol{z}, \boldsymbol{d}\}$. Using the form of $Q_\psi$ in Eq 7, this likelihood can be decomposed by positions. Thus we can conveniently train a bi-RNN which maximize the condition likelihood of latent variables by stochastic gradient descent.

**Remark:** We can get multiple samples $\{\boldsymbol{z}, \boldsymbol{d}\}$ for each $\boldsymbol{x}$ from $\tilde{Q}(\boldsymbol{z}, \boldsymbol{d}|\boldsymbol{x})$ to reduce the variance in stochastic gradient. In our algorithm, the samples of latent variable come naturally from the auxiliary distributions (which are integrated with penalty method), rather than the derivation from lower bound of objective (Tang & Salakhutdinov, 2013; Raiko et al., 2014; Mnih & Rezende, 2016).

## 5 EXPERIMENTS

**Baselines** We compare with classical HSMM and two popular HSMM variants. The first one is Hierarchical Dirichlet-Process HSMM (HDP-HSMM) (Johnson & Willsky, 2013), which is the nonparametric Bayesian extension to the traditional HSMM that allows infinite number of hidden states; the second one is called subHSMM (Johnson & Willsky, 2014), which uses infinite HMM as the emission model for each segment. This model also has two-level of latent structure. It considers the dependency within each segment, which is a stronger algorithm than HDP-HSMM. We also compare with the CRF autoencoder (CRF-AE) (Ammar et al., 2014), which uses markovian CRF as recognition model and conditional *i.i.d.* model for reconstruction. Comparing to HSMM, this model ignores the segmentation structures in modeling and is more similar to HMM.

**Evaluation Metric** We evaluate the performance of each method via the labeling accuracy. Specifically, we compare the labels of each single observations in each testing sequence. Since the labels are unknown during training, we use KM algorithm (Munkres, 1957) to find the best mapping between predicted labels and ground-truth labels.

**Settings** Without explicitly mentioned, we use leave-one-sequence-out protocol to evaluate the methods. Each time we test on one held-out sequence, and train on other sequences. We report the mean accuracy in Table 1. We set the truncation of max possible duration $D$ to be 400 for all tasks. We also set the number of hidden states $K$ to be the same as ground truth.

For the HDP-HSMM and subHSMM, the observation distributions are initialized as standard Multivariate Gaussian distributions. The duration is modeled by the Poisson distribution. We tune the concentration parameters $\alpha, \gamma \in \{0.1, 1, 3, 6, 10\}$. The hyperparameters are learned automatically. For subHSMM, we tune the truncation threshold of the second level infinite HMM from $\{2 \ldots 15\}$.

For CRF-AE, we extend the origin model for the continuous observations, and learn all parameters similar to M. Schmidt (2008). We use mixture of Gaussians to model the emission, where the number of mixtures is tuned in $\{1, \ldots, 10\}$.

For the proposed R-HSMM, we use Adam (Kingma & Ba, 2014) to train the $K$ generative RNN and bi-RNN encoder. To make the learning tractable for long sequences, we use back propagation through time (BPTT) with limited budget. We also tune the dimension of hidden vector in RNN, the $L_2$-regularization weights and the stepsize. We implemented with CUDA that parallelized for different RNNs, and conduct experiments on K-20 enabled cluster. We include both the R-HSMM with the exact MAP via dynamic programming (rHSMM-dp) and sequential VAE with forward pass (rHSMM-fw) in experiments. In all tasks, the rHSMM-fw achieves almost the same performance to rHSMM-dp, but 400 times faster, showing the bi-RNN is able to mimic the forward-backward algorithm very well with efficient computation.

### 5.1 SEGMENTATION ACCURACY

**Synthetic Experiments** We first evaluate the proposed method on two 1D synthetic sequential data sets. The first data set is generated by a HSMM with 3 hidden states, where $\pi, A, B$ are designed beforehand. A segment with hidden state $z$ is a sine function $\lambda_z \sin(\omega_z x + \epsilon_1) + \epsilon_2$, where $\epsilon_1$ and $\epsilon_2$ are Gaussian random noises. Different hidden states use different scale parameters $\lambda_z$ and frequency parameters $\omega_z$. The second data set also has 3 hidden states, where the segment with hidden state $z$ is sampled from a Gaussian process (GP) with kernel function $k_z(x, y)$. Different hidden states employ different kernel functions. The specific kernel functions used here are $k_1(x, y) = \exp\{-\min(|x-y|, |x+y|)^2/10\}$, $k_2(x, y) = \exp\{-(x-y)^2/10\}$ and $k_3(x, y) = (5 - |x-y|)I\{(5-|x-y|) < 5\}$. For both of the Sine and GP data sets, the duration of a segment is randomly sampled from a distribution defined on $\{1, ..., 100\}$, which depends on the hidden states. Thus, the segmentation task corresponds to finding out different functions embedded in the sequences.

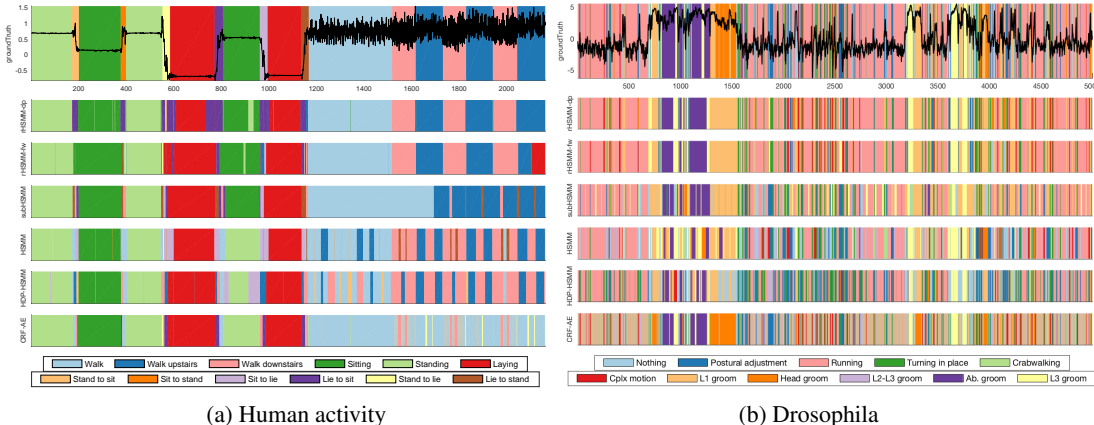

(a) Human activity

(b) Drosophila

Figure 3: Segmentation results on Human activity and Drosophila datasets. Different background colors represent the segmentations with different labels. In the top row, the black cure shows the signal sequence projected to the first principle component. The following two rows are our algorithms which almost exact locate every segment. (a) The Human activity data set contains 12 hidden states, each of which corresponds to a human action; (b) The Drosophila data set contains 11 hidden states, each of which corresponds to a drosophila action.

Table 1: Error rate of segmentation. We report the mean and standard deviation of error rate.

| Methods | SINE | GP | HAPT | Drosophila | Heart | PN-Full |
|---|---|---|---|---|---|---|
| rHSMM-dp | **2.67 ± 1.13%** | **12.46 ± 2.79%** | **16.38 ± 5.03%** | **36.21 ± 1.37%** | **33.14 ± 7.87%** | **31.95 ± 4.32%** |
| rHSMM-fw | 4.02 ± 1.37% | **13.13 ± 2.89%** | **17.74 ± 7.64%** | **35.79 ± 0.51%** | **33.36 ± 8.10%** | **32.34 ± 3.97%** |
| HSMM | 41.85 ± 2.38% | 41.15 ± 1.99% | 41.59 ± 8.58% | 47.37 ± 0.27% | 50.62 ± 4.20 % | 45.04 ± 1.87% |
| subHSMM | 18.14 ± 2.63% | 24.81 ± 4.63% | 22.18 ± 4.45% | 39.70 ± 2.21% | 46.67 ± 4.22% | 43.01 ± 2.35% |
| HDP-HSMM | 42.74 ± 2.73% | 41.90 ± 1.58% | 35.46 ± 6.19% | 43.59 ± 1.58% | 47.56 ± 4.31% | 42.58 ± 1.54% |
| CRF-AE | 44.87 ± 1.63% | 51.43 ± 2.14% | 49.26 ± 10.63% | 57.62 ± 0.22% | 53.16 ± 4.78% | 45.73 ± 0.66% |

We visualize the segmentation results of ground truth and three competitors on Sine and GP data sets in Figure 1a and Figure 1b respectively, and report the numerical results in Table 1. As we can see, R-HSMM provides much better results on even small segments, dramatically outperforms HSMM variants and CRF-AE. Also note that, the sine function depicts short term dependencies, while Gaussian process has long dependency that determined by the kernel bandwidth. This demonstrates the ability of R-HSMM in capturing the long or short term dependencies.

**Human activity** This dataset which is collected by Reyes-Ortiz et al. (2016) consists of signals collected from waist-mounted smartphone with accelerometers and gyroscopes. Each of the volunteers is asked to perform a protocol of activities composed of 12 activities (see Figure 3a for the details). Since the signals within an activity type exhibit high correlation, it is natural for RNN to model this dependency. We use these 61 sequences, where each sequence has length around 3000. Each observation is a 6 dimensional vector, consists of triaxial measures from accelerometers and gyroscopes.

Figure 3a shows the ground truth and the segmentation results of all methods. Both rHSMM-dp and rHSMM-fw almost perfectly recover the true segmentation. They can also capture the transition activity types, *e.g.*, stand to lie or sit to lie. The HSMM, HDP-HSMM and CRF-AE makes some fragmental but periodical segmentations for walking, caused by lacking the dependency modeling within a segment. The subHSMM also has similar problem, possibly due to the limited ability of HMM generative model.

**Drosophila** Here we study the behavior patterns of drosophilas. The data was collected by Kain et al. (2013) with two dyes, two cameras and some optics to track each leg of a spontaneously behaving fruit fly. The dimension of observation in each timestamp is 45, which consists of the raw features and some higher order features. See Figure 3b for the detail of the 11 behavior types. We perform leave-one-sequence-out experiment on 10 sequences of length 10000 each. Figure 3b shows the segmentation results on the prefix of one sequence, while Table 1 gives the mean accuracy on all sequences. Different from the previous experiment, where the human activity signals are relatively

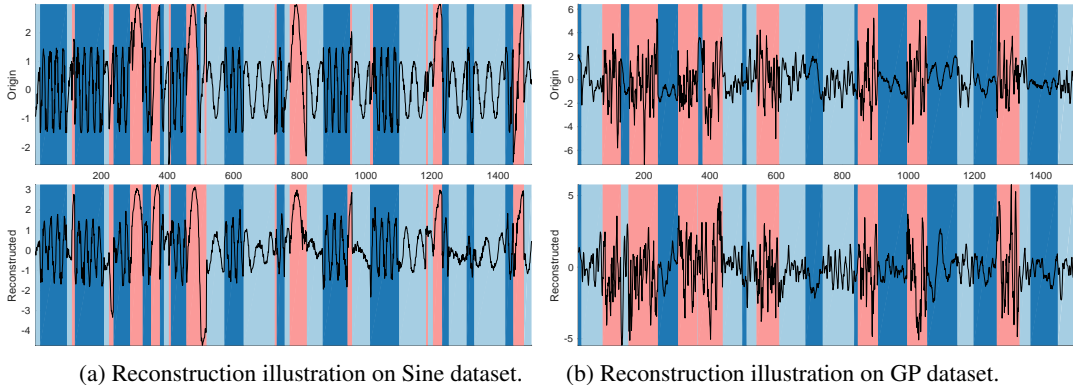

(a) Reconstruction illustration on Sine dataset.  (b) Reconstruction illustration on GP dataset.

Figure 4: Reconstruction illustration. The generative RNNs (decoders) are asked to reconstruct the signals from only the discrete labels and durations (which are generated from encoder).

smooth, here the signals depict high variance. Different activities exhibit quite different duration and patterns. Also, the activity types changes frequently. The R-HSMM almost captured each changing point of activities with both long and short durations. The corresponding mean accuracy also outperforms the baselines. However, we observed there are some correct segmentations with wrong labels. This happens mostly to the short segments, in which the RNN doesn't have enough history established for distinguishing similar activity types.

**Physionet**   The heart sound records, usually represented graphically by phonocardiogram (PCG), are key resources for pathology classification of patients. We collect data from PhysioNet Challenge 2016 (Springer et al., 2015), where each observation has been labeled with one of the four states, namely Diastole, S1, Systole and S2. We experiment with both the raw signals and the signals after feature extraction. Regarding the raw signals (Heart dataset), we collect 7 1-dimensional sequences of length around $40000$. The feature-rich dataset (PN-Full) contains 2750 sequences, where each of them consists of 1500 4-dimensional observations. We do 5-fold cross validation for PN-Full. The visualization of segmentation results are shown in Appendix B.4. As the results shown in Table 1, our algorithm still outperforms the baselines significantly. Also for such long raw signal sequences, the speed advantage of bi-RNN encoder over Viterbi is more significant. Viterbi takes 8min to do one inference, while bi-RNN only takes several seconds. Our framework is also flexible to incorporate prior knowledge, like the regularity of heart state transition into HSMM.

## 5.2   RECONSTRUCTION

In this section, we examine the ability of learned generative model by visualizing the reconstructed signals. Given a sequence $x$, we use recognition model to get the latent variables $z$ and $d$, then use learned $K$ generative RNNs to generate signals within each segment. For the ease of visualization, we show the results on 1D signal dataset in Fig. 4a and Fig. 4b.

From Fig. 4 we can see the generative RNN correctly captures different characteristics from signals of different segment labels, such as different frequencies and scales in Sine dataset, or the different variance patterns in GP dataset. This is essential to distinguish between different segments.

## 6   CONCLUSION

We presented the R-HSMM, a generalization of HSMM by incorporating recurrent neural generative model as the emission probability. To eliminate the difficulty caused by such flexible and powerful model in inference, we introduced the bi-RNN as the encoding distribution via the variational autoencoder framework to mimic the forward-backward algorithm. To deal with the difficulty of training VAE containing discrete latent variables, we proposed a novel stochastic distributional penalty method. We justified the modeling power of the proposed R-HSMM via segmentation accuracy and reconstruction visualization. From the comprehensive comparison, the proposed model significantly outperforms the existing models. It should be emphasized that the structured bi-RNN encoder yields similar performance as the exact MAP inference, while being 400 times faster. Future work includes further speeding up of our algorithm, as well as generalizing our learning algorithm to other discrete variational autoencoder.

ACKNOWLEDGMENTS

This project was supported in part by NSF IIS-1218749, NIH BIGDATA 1R01GM108341, NSF CAREER IIS-1350983, NSF IIS-1639792 EAGER, ONR N00014-15-1-2340, Nvidia and Intel.

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

# Appendix

## A  OPTIMIZING DYNAMIC PROGRAMMING

### A.1  SQUEEZE THE MEMORY REQUIREMENT

In this section, we show that the Eq. 13 can be computed in a memory efficient way. Specifically, the dynamic programming procedure can be done with $\mathcal{O}(|\boldsymbol{x}|K)$ memory requirement, and caching for precomputed emission probabilities requires $\mathcal{O}(D^2 K)$ memory space.

**Update forward variable** $\alpha$   Note that in Eq. 13, when $r > 1$, we can update $\alpha_t(j, r)$ deterministically. So it is not necessary to keep the records for $r > 1$.

Specifically, let's only record $\alpha_t(j, 1)$, and do the updates in a similar way as in Eq. 13. The only difference is that, when constructing the answer, *i.e.*, the last segment solution, we need to do a loop over all possible $z$ and $d$ in order to find the best overall segmentation solution.

It is easy to see that the memory consumption is $\mathcal{O}(|\boldsymbol{x}|K)$.

**Caching emission probability**   At each time step $t$, we compute $P(x_{t+r}|x_{t:t+r-1}, z = j)$ for each $j \in \mathbb{Z}$ and $r \in \mathbb{D}$. That is to say, we compute all the emission probabilities of observations starting from time $t$, and within max possible duration $D$. This can be done by performing feed-forward of $K$ RNNs. After that, storing these results will require $\mathcal{O}(KD)$ space. For simplicity, we let $e^t_{j,r} = P(x_{t+r}|x_{t:t+r-1}, z = j)$, where $e^t \in \mathbb{R}^{K \times D}$.

Note that, at a certain time step $t$, we would require the emission probability of observations $P(x_t|x_{t-r+1:t-1}, z = j)$ for some $j \in \mathbb{Z}$ and $r \in \mathbb{D}$. In this case, the corresponding first observation is $x_{t-r}$. That is to say, we should keep $e^{t-D+1}, \ldots, e^t$ at time step $t$. This makes the memory consumption goes to $\mathcal{O}(KD^2)$

### A.2  SQUEEZE THE TIME COMPLEXITY

In Eq. 13, the most expensive part is when $r = 1$ and $t > 1$. If we solve this in a naive way, then this step would require $\mathcal{O}(|\boldsymbol{x}|K^2 D)$ for time complexity, which is quite expensive.

Here we adopt similar technique as in Yu & Kobayashi (2003). Let $\gamma_t(i) = \max_{r' \in \mathbb{D}} \alpha_{t-1}(i, r')$, then we can get

$$\alpha_t(j, r) = \max_{i \in \mathbb{Z}} \max_{r' \in \mathbb{D}} \alpha_{t_1}(i, r') + \frac{1}{1+\lambda} \log(A_{i,j} B_{j,1} P(x_t|z = j)) \tag{17}$$

$$+ \frac{\lambda}{1+\lambda} \log Q_\psi(z_{t-r+1} = j, d_{t-r+1} = r|\boldsymbol{x})$$

$$= \max_{i \in \mathbb{Z}} \gamma_{t-1}(i) + \frac{1}{1+\lambda} \log(A_{i,j} B_{j,1} P(x_t|z = j)) \tag{18}$$

$$+ \frac{\lambda}{1+\lambda} \log Q_\psi(z_{t-r+1} = j, d_{t-r+1} = r|\boldsymbol{x})$$

This reduces the complexity to be $\mathcal{O}(|\boldsymbol{x}|K^2)$.

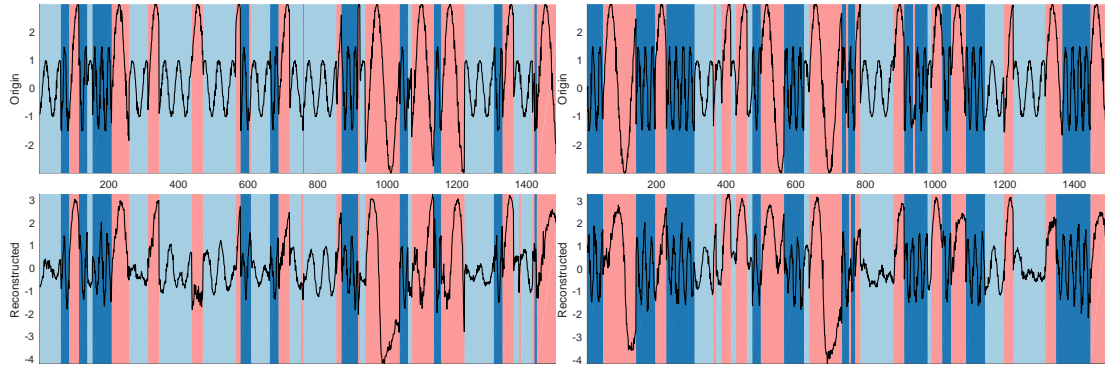

Figure 5: More reconstruction illustration on Sine dataset.

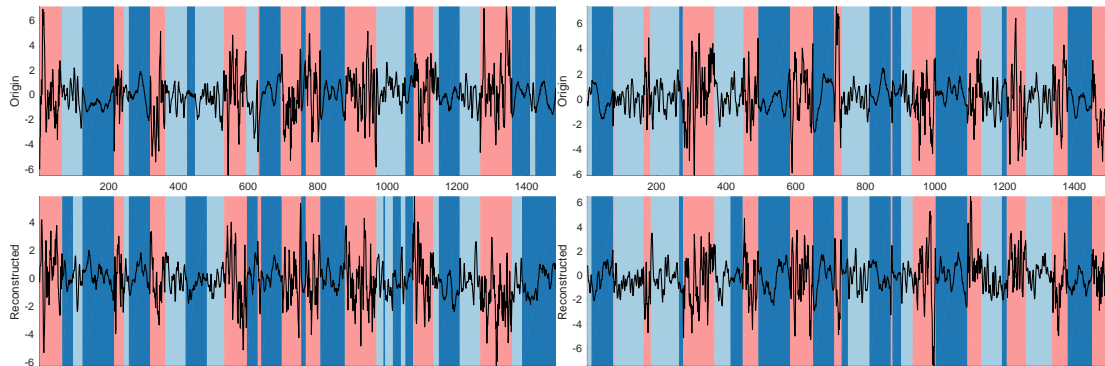

Figure 6: More reconstruction illustration on Gaussian Process dataset.

## B    MORE EXPERIMENT RESULTS

### B.1    SYNTHETIC DATASETS

The reconstructed signals from the original signals are shown in Fig. 5 and Fig. 6 for sine dataset and gaussian Process dataset respectively. We can see the reconstructed signal almost recovered the original signal. The RNN captured the key differences of states, such as the frequency and scale; while in gaussian process dataset, it also recovered the complicated pattern involving long term dependencies.

We show the confusion matrix of all methods on synthetic sine and gaussian process dataset in Figure 7 and Figure 8 respectively.

### B.2    HUMAN ACTIVITY

The confusion matrices of our method and two baseline algorithms are shown in Figure 9.

In Figure 10, we also show several other segmentation results on different testing sequences.

### B.3    DROSOPHILA

The confusion matrices of our method and two baseline algorithms are shown in Figure 11.

Since each sequence is too long to be clearly shown in one figure, we split the segmentation results of one sequence into four parts, and show them in Figure 12.

### B.4    HEART SOUND RECORDS

The confusion matrices of our method and two baseline algorithms are shown in Figure 13.

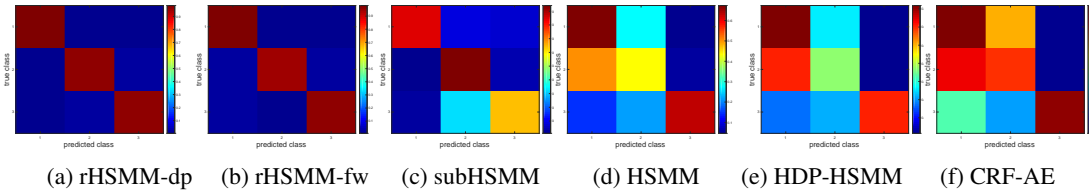

(a) rHSMM-dp (b) rHSMM-fw (c) subHSMM (d) HSMM (e) HDP-HSMM (f) CRF-AE

Figure 7: Confusion matrix on Synthetic Sine dataset.

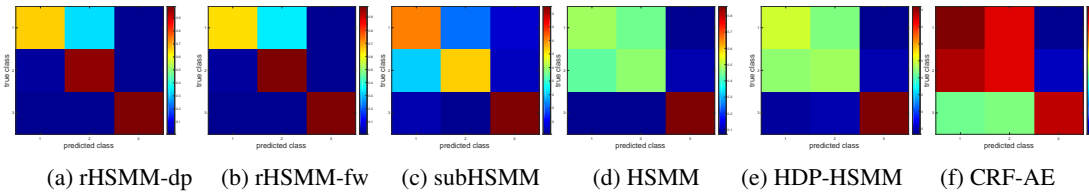

(a) rHSMM-dp (b) rHSMM-fw (c) subHSMM (d) HSMM (e) HDP-HSMM (f) CRF-AE

Figure 8: Confusion matrix on Synthetic Gaussian Process dataset.

Also, we split the segmentation results of one sequence into four parts, and show them in Figure 14.

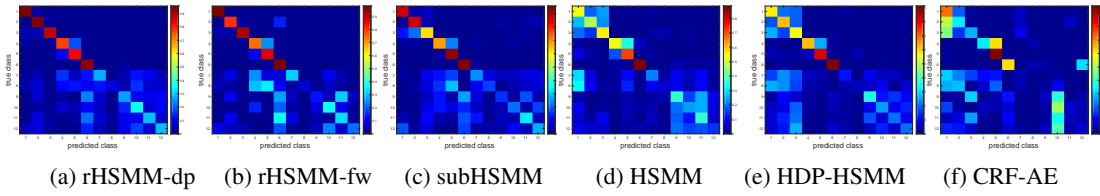

(a) rHSMM-dp   (b) rHSMM-fw   (c) subHSMM   (d) HSMM   (e) HDP-HSMM   (f) CRF-AE

Figure 9: Confusion matrix on Human Activity dataset.

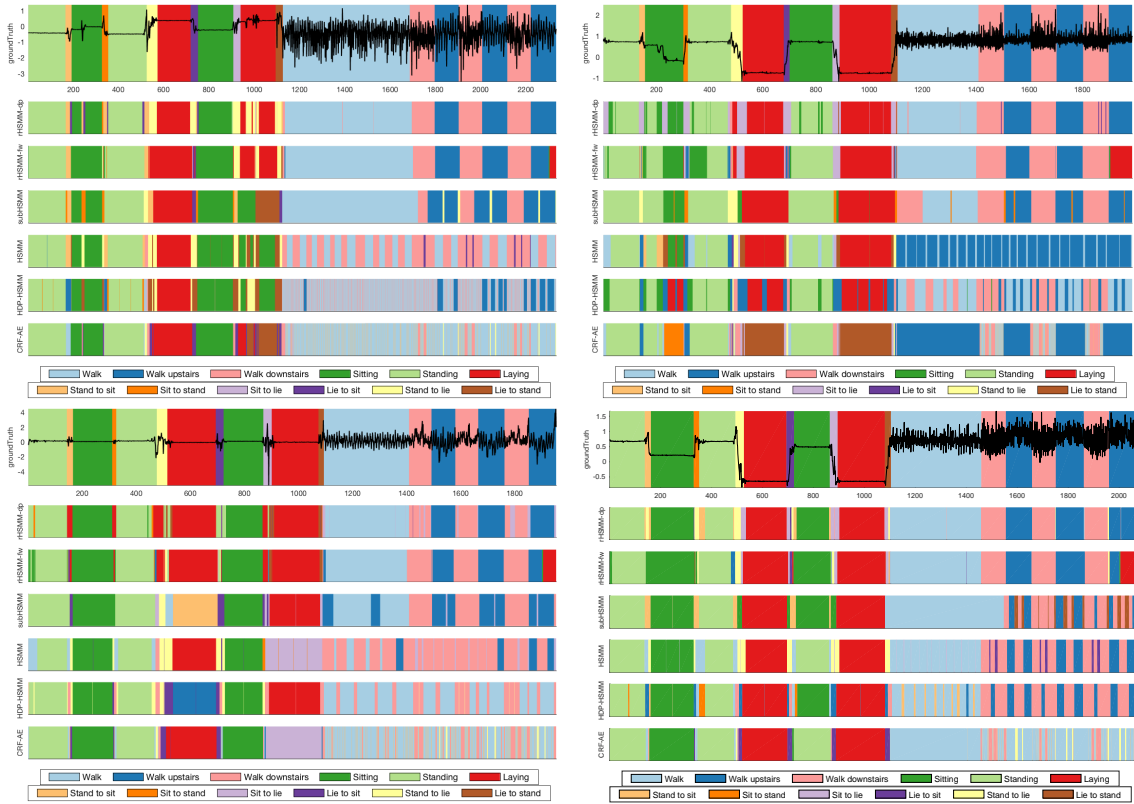

Figure 10: More segmentation results on Human Activity dataset.

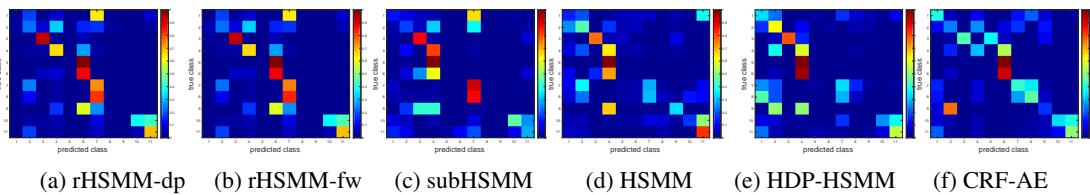

(a) rHSMM-dp   (b) rHSMM-fw   (c) subHSMM   (d) HSMM   (e) HDP-HSMM   (f) CRF-AE

Figure 11: Confusion matrix on Drosophila dataset.

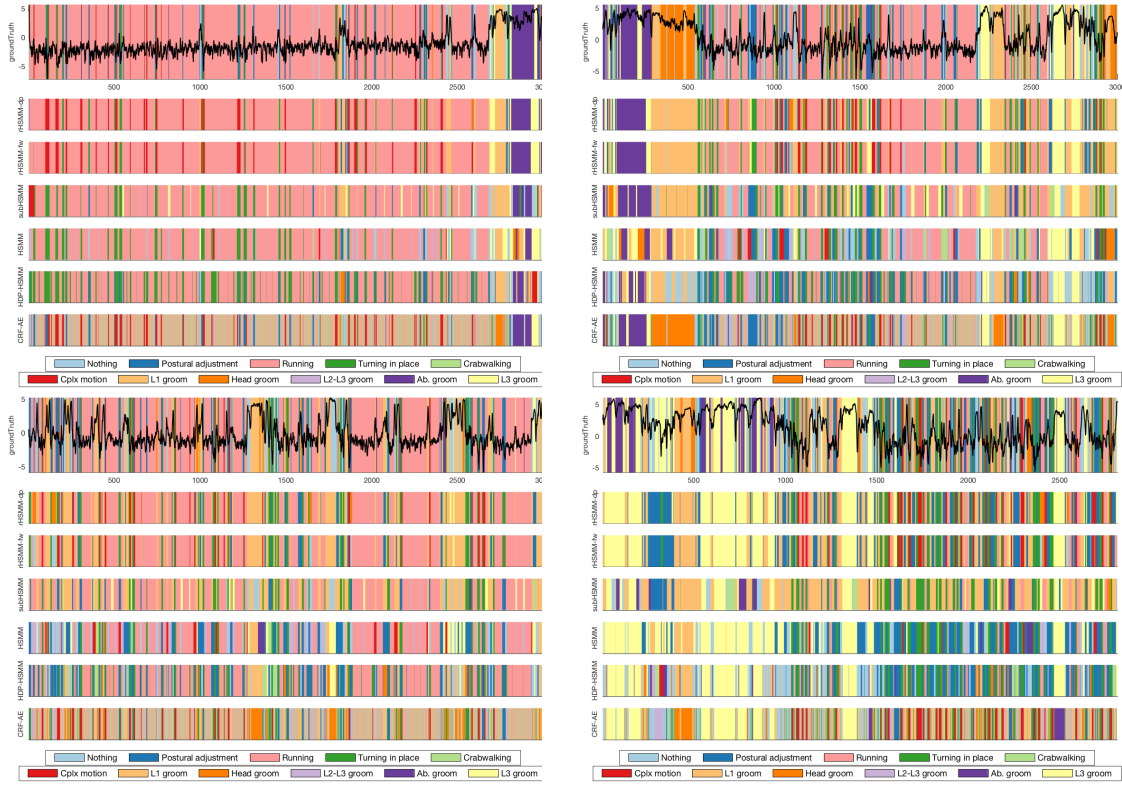

Figure 12: More segmentation results on Drosophila dataset.

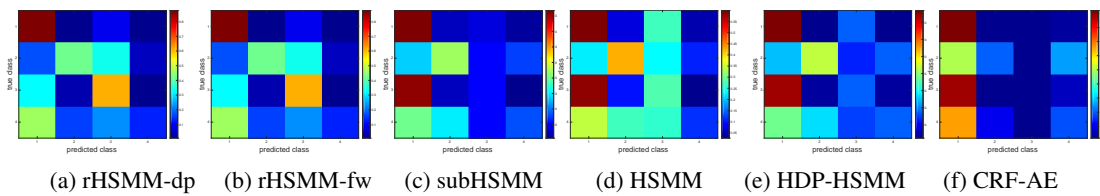

(a) rHSMM-dp  (b) rHSMM-fw  (c) subHSMM  (d) HSMM  (e) HDP-HSMM  (f) CRF-AE

Figure 13: Confusion matrix on Heart Sound dataset.

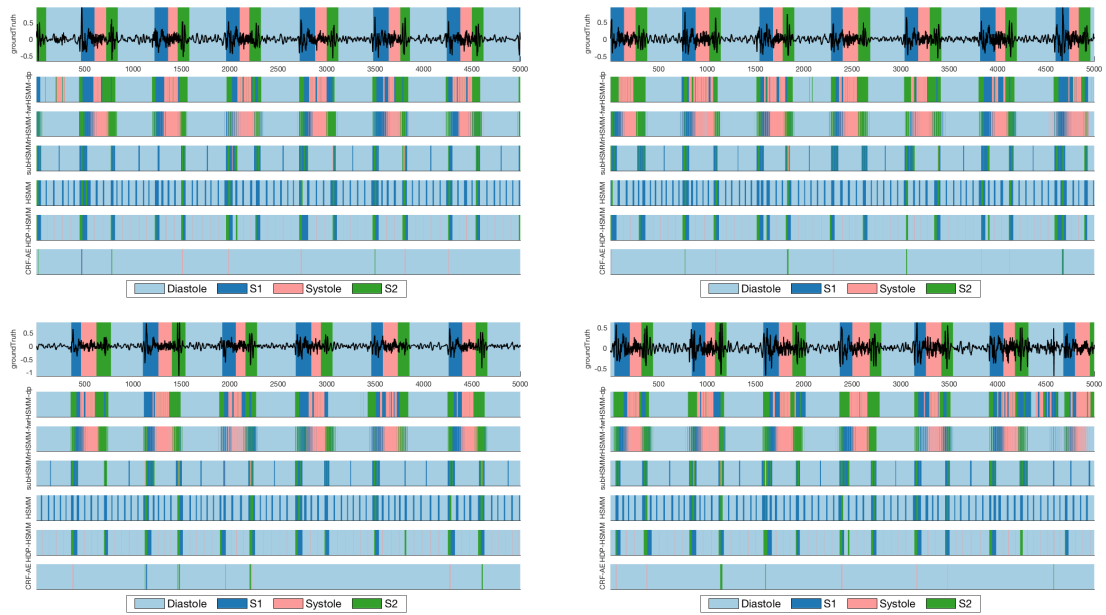

Figure 14: More segmentation results on Heart Sound dataset.

