# Peer review of "Recurrent Hidden Semi-Markov Model"

_ICLR 2017 — accepted_

[Official Review · AnonReviewer2 · rating 7 · confidence 4 · 10 Dec 2016]
**Novel model for temporal data**
substance 3

This paper presents a novel model for unsupervised segmentation and classification of time series data.  A recurrent hidden semi-markov model is proposed.  This extends regular hidden semi-markov models to include a recurrent neural network (RNN) for observations.  Each latent class has its own RNN for modeling observations for that category.  Further, an efficient training procedure based on a variational approximation.  Experiments demonstrate the effectiveness of the approach for modeling synthetic and real time series data.

This is an interesting and novel paper.  The proposed method is a well-motivated combination of duration modeling HMMs with state of the art observation models based on RNNs.  The combination alleviates shortcomings of standard HSMM variants in terms of the simplicity of the emission probability.  The method is technically sound and demonstrated to be effective.

It would be interesting to see how this method compares quantitatively against CRF-based methods (e.g. Ammar, Dyer, and Smith NIPS 2014).  CRFs can model more complex data likelihoods, though as noted in the response phase there are still limitations.  Regardless, I think the merits of using RNNs for the class-specific generative models are clear.

[Official Review · AnonReviewer3 · rating 7 · confidence 3 · 17 Dec 2016]
substance 5

Putting the score for now, will post the full review tomorrow.

[Official Review · AnonReviewer1 · rating 7 · confidence 4 · 21 Dec 2016]
**Good method for HSMM estimation**
soundness 3

This paper proposes a novel and interesting way to tackle the difficulties of performing inference atop HSMM. The idea of using an embedded bi-RNN to approximate the posterior is a reasonable and clever idea. 

That being said, I think two aspects may need further improvement:
(1) An explanation as to why a bi-RNN can provide more accurate approximations than other modeling choices (e.g. structured mean field that uses a sequential model to formulate the variational distribution) is needed. I think it would make the paper stronger if the authors can explain in an intuitive way why this modeling choice is better than some other natural choices (in addition to empirical verification).
(2) The real world datasets seem to be quite small (e.g. less than 100 sequences). Experimental results reported on larger datasets may also strengthen the paper.

[Author Response · Hanjun Dai · 12 Jan 2017]
**Paper revision**

Dear reviewers, we have revised our paper according to your insightful suggestions and comments.

Specifically, we added a baseline CRF-autoencoder, and did the quantitative comparison across all datasets. We’ve also added a large dataset, which contains 2750 sequences, where each sequence has ~1.5k 4-D observations.

For more details about the dataset (called PN-FULL in the paper) and comparison with new baseline (called CRF-AE in the paper),  please check our revised pdf.

[Final Decision · Program Chairs · 06 Feb 2017]
**ICLR committee final decision**

This paper is a by-the-numbers extension of the hidden semi-Markov model to include nonlinear observations, and neural network-based inference. The paper is fairly clear, although the English isn't great. The experiments are thorough.
 
 Where this paper really falls down is on originality. In particular, in the last two years there have been related works that aren't cited (and unfortunately weren't mentioned by the reviewers) that produce similar models. In particular, Johnson et al's 2016 NIPS paper develops almost the same inference strategy in almost the same model class.